# Interactive Impact of Sleep Duration and Sleep Quality on the Risk of Developing Metabolic Syndrome in Korean Adults

**DOI:** 10.3390/healthcare8020186

**Published:** 2020-06-25

**Authors:** Young-Joo Kim, Hyun-E Yeom

**Affiliations:** 1Department of Economics, Hongik University, Seoul 04066, Korea; y.j.kim@hongik.ac.kr; 2College of Nursing, Chungnam National University, Daejeon 35015, Korea

**Keywords:** differential threshold, healthy lifestyle, metabolic syndrome, sleep

## Abstract

Sleep quality is important for the normal functioning of hormonal and metabolic processes in the body; however, few studies have considered the effects of both sleep duration and sleep quality on predicting metabolic syndrome risk. We examined the interactive impact of sleep duration and sleep quality on the risk of developing metabolic syndrome using logistic regression analysis with a threshold based on hours of sleep. Data were collected from 411 adults in South Korea and, according to the estimated threshold of 6 h of sleep (95% Confidence Interval, CI = 5–7 h), participants were classified as short (<6 h) or adequate-long (≥6 h) sleepers. The two groups differed significantly on various health measures. While short sleepers were more likely than adequate-long sleepers to experience adverse health conditions, which increased their risk of developing metabolic syndrome, they were not influenced by sleep quality. For adequate-long sleepers, however, a decrease in sleep quality was associated with an increased risk of developing metabolic syndrome (odds ratio = 1.24, 95% CI = 1.07–1.43). Our results suggest that both sleep duration and sleep quality are crucial determinants of the development of metabolic syndrome and that it is important to maintain at least 6 h of sleep.

## 1. Introduction

Metabolic syndrome (MetS) represents a cluster of risk factors, including central obesity, high fasting blood glucose levels, and hypertension, which raises the chances of developing cardiovascular diseases (CVDs) and other health concerns. With a rise in prevalence, MetS is gaining global attention due to the fact that it increases mortality and also threatens physical and psychological aspects of an individual’s quality of life [1].

Sleep is one of the key determinants of an individual’s quality of life as it affects health through its interaction with metabolic mechanisms entangled with the sympathetic nervous system (SNS) [2]. According to several studies [3,4], sleep deprivation causes obesity by influencing appetite while also elevating the risk of diabetes by raising both glucose tolerance and insulin resistance. Further, insufficient sleep activates the SNS (thereby elevating blood pressure and blood cortisol concentration), weakens the immune system [5], affects cognitive and psychological states, such as memory and anxiety [6], and behavioral characteristics, including physical activity and dietary habits [3]. Given the evidence of adverse effects of sleep debt and sleep disorders [7], identifying crucial characteristics of sleep that are associated with the risk of developing MetS becomes critical in the prevention and delay of MetS.

In the existing literature, optimal sleep duration among generally healthy individuals is considered to be between 7 and 8 h per day [1,8]. Many studies have demonstrated that short sleepers (<6 h of sleep per day) and long sleepers (≥9 h of sleep per day) show elevations in various health risk factors compared to adequate sleepers (7–8 h of sleep per day) [9] and that there is a U-shaped relationship between sleep duration and health risk [7,10].

However, there is disagreement regarding the relationship between sleep duration and the risk of having MetS. A meta-analysis [11] reported that short sleep duration is a stronger risk factor for developing MetS than long sleep duration, but other studies found that either short or long sleep is associated with the risk for MetS [12,13]. These inconsistent results may be due to the fact that sleep is multi-faceted and thus risk factors for MetS are not limited to sleep duration but also associated with the quality of sleep. For example, some studies have reported that both sleep duration and sleep quality are predictors of CVDs such as diabetes and hypertension [4], whereas other studies reported that sleep quality is a stronger predictor than sleep duration [14]. When considering the relationship between sleep quality and sleep duration, quality tends to deteriorate with increasing duration [15], though any duration that deviates from the optimal duration may worsen sleep quality [12,13]. Furthermore, it has been argued that sleep quality is more closely associated with sleep regularity than duration [16]. Although sleep quality has been documented to be an important component for normal functioning of hormonal and metabolic processes in the body, few studies have considered the effects of both sleep duration and sleep quality in predicting MetS risk. While the risk of MetS is compounded by demographic (e.g., age, income, shift work) [17] and behavioral characteristics (e.g., nutrition, physical activity, alcohol intake, smoking) [18], few studies have taken into consideration demographic and behavioral characteristics when assessing the effects of sleep on the development of MetS.

In this study, we investigated the interactive effect of sleep quality and sleep duration on the risk of developing MetS, controlling for risk factors that are related to sociodemographic and health behaviors, to better understand the impact of sleep on the incidence of MetS. As a novel approach, we applied a threshold regression to identify different patterns in the impact of sleep quality depending on a specific point of sleep duration in the estimation of the MetS risk. By considering the interactive effect of sleep quality and sleep duration by means of the threshold regression, we aimed to accurately capture the role of sleep in predicting MetS risk.

## 2. Materials and Methods

### 2.1. Participants

This was a cross-sectional study of 411 participants recruited from outpatient clinics at three general hospitals and community centers in Daegu and Gyeongju, cities of South Korea. We identified participants as a risk group of MetS if they met two or more of the risk factor criteria defined by the National Cholesterol Education Program’s Adult Treatment Panel III [19] guidelines, combined with the World Health Organization’s (WHO) criteria for central obesity in Asian populations [20]. The five risk factors were as follows: (1) high blood pressure, defined as ≥130 mmHg systolic or ≥85 mmHg diastolic, or taking medication for managing hypertension, (2) hyperglycemia, defined as fasting blood glucose ≥100 mg/dL, or taking medication for treatment, (3) hypertriglyceridemia, defined as fasting plasma triglycerides ≥150 mg/dL, or taking medication for treatment, (4) low-density lipoprotein cholesterol <40 mg/dL in men, <50 mg/dL in women, or taking medication for treatment, and (5) central obesity, defined as an abdominal circumference ≥90 cm in men and ≥ 85 cm in women.

A prior power analysis was conducted to estimate an adequate sample size using G * Power 3.1.9.4 software (Heinrich-Heine-University, Düsseldorf, Germany) [21]. Using the type 1 error (α) criterion of 0.05, medium effective size (0.20) for linear multiple regression analysis with 15 predictors, and anticipated power level (0.08), the minimum required sample size was calculated to be 108. When considering two groups categorized by threshold point of sleep duration, a minimum number of 216 participants was required. Since the final total number of the participants in the present study was 411 and both groups on the threshold criterion consisted of over 108 participants, the sample size of the present study was adequate.

### 2.2. Measures of Sleep Quality and Duration

The quality of sleep was assessed with the Korean version of the Pittsburgh Sleep Quality Index (PSQI) [22]. The Pittsburgh Sleep Quality Index (PSQI) is a scale used globally in a variety of populations to assess overall quality of sleep [23]. The PSQI consists of 19 items and measures seven domains of sleep: subjective sleep quality, sleep latency, sleep duration, habitual sleep efficiency, sleep disturbance, use of sleep medication, and daytime dysfunction. Each item is assessed on a 4-point Likert scale ranging from 0 to 3, and the total score of sleep quality is calculated based on the algorithm described in the PSQI [23], with higher scores indicating poorer quality of sleep. Sleep duration was assessed with a question that asked the average number of hours they slept each night during the past month, “how many hours of actual sleep did you get each night, on average?” The reliability of the items assessing sleep quality from this study was Cronbach’s alpha = 0.86.

### 2.3. Covariates

We considered sociodemographic and health-related lifestyle characteristics as covariates in the analyses. Sociodemographic information was collected via a questionnaire that asked about participants’ age, gender, education, income level, and marital and living status. Health-related behaviors were obtained using the 25-item Health Behavior Scale, which is a validated index for the prevention of CVDs [24]. The Health Behavior Scale asks questions about different health behaviors, such as engagement in physical activity, diet habits, and health responsibility, and responses are reported on a 4-point Likert scale indicating the frequency of the behavior. A higher total score indicates better health-related behavior. In addition, information on whether the participant was a current smoker (yes or no) or a current drinker (yes or no) were collected. The participant’s height and body weight were assessed to calculate their body mass index (BMI) using the formula (kg/m^2^). Lastly, experience of symptoms was assessed with a 20-item symptom checklist (yes, no) that consisted of cardiovascular symptoms (e.g., shortness of breath, swelling in legs or face) and general physical symptoms (e.g., indigestion, tingling, and fatigue) prevalent in the adult Korean population. The total number of symptoms was calculated.

### 2.4. Ethical Consideration and Procedure

All procedures of this study were approved by the Institutional Review Board of the university where the corresponding author was affiliated (110757-201605-HR-05-05, 201711-SB-081-01). All potential participants were provided information about the protection of participant’s rights, the physical and psychological well-being of human participants, and the study purpose and procedure. If they agreed to participate in the study after being informed, participants filled out a written informed consent and the self-administered surveys.

### 2.5. Statistical Analyses

All statistical analyses were performed using Stata version 14 (Stata Corporation, College Station, TX, USA). Descriptive statistics were computed to summarize basic quantitative characteristics of all variables for this study. Preliminary analyses were performed to describe key features of the data and prepare further analyses.

We employed a threshold regression model to assess the potential asymmetric interaction between sleep quality and sleep duration on the development of MetS. The concept of threshold effect has been widely reported in various fields, from biology to toxicology to the medical literature, to explain heterogeneous treatment effects due to a covariate. For example, Kamper-Jorgensen and colleagues [25] have reported the threshold effect of alcohol intake and cirrhosis, and Hill et al. [26] reported on the threshold effect of exercise intensity and circulating cortisol level. The threshold for sleep duration was determined to maximize the joint logit-likelihood function as in the study by Lee et al. [27]. Precise construction of confidence intervals for the threshold of sleep duration was challenging because hours of sleep, which we will refer to as “sleep duration,” was measured in 30-min increments in this study and, therefore, was not continuous. Against this backdrop, we report a confidence interval for the threshold sleep duration, as in the study by Hansen [28].

We next split our cohort into two groups, depending on whether sleep duration was above or below the threshold, and conducted t-tests and chi-square tests to examine whether there were differences in sociodemographic and health-related characteristics between the subgroups. Pearson’s correlation coefficients were applied to assess the relationship between sleep quality and sleep duration.

We used logistic regression analyses to examine the effects of various covariate factors, not limited to sleep characteristics, on the risk of MetS for each subgroup divided by the threshold. We provided the estimated coefficients of various risk factors from logistic regression analyses first in the full cohort and then in each of the two subgroups, above or below the threshold, respectively.

## 3. Results

### 3.1. Descriptive Characteristics of Participants

Our study cohort consisted of 411 adults, 53% of whom were men. They ranged in age from 25 to 84, with a mean age of 53. Half of the cohort (51.6%) had obtained a college education and 76% reported having a job. Income levels (average monthly income) were divided into five categories, from 1 (below 1 million Korean won or 830 USD) through 5 (above 4 million Korean won or 3300 USD), with the average income level being 3, which is analogous to 2.5 million Korean won or 2075 USD.

The average BMI was 24.6 and about 60% of the cohort was at risk for MetS. The average number of symptoms was 10.34 (standard deviation (SD) = 5.68, range = 0–20) and the mean score of health behavior was 2.42 (SD = 0.48, range = 0–3). The proportion of current smokers and drinkers was 23.4% and 47.0%, respectively.

Figure 1 presents the odds ratio of the MetS risk related to daily sleep duration. The odds ratio was higher at the two tails of the sleep duration distribution, and lower toward the center of the distribution. This variation in odds ratio along the line of sleep duration led us to investigate a potential heterogeneity in the incidence of MetS depending on sleep duration.

### 3.2. Threshold of Sleep Duration

In the estimation of the threshold point, we controlled for individual characteristics including age, gender, education, and other variables that are listed in Table 1. The estimated threshold point of sleep duration was 6 h (95% CI = 5–7 h). Figure 2 plots the likelihood ratio sequence along candidate thresholds. The confidence interval for the threshold is the collection of values below the dashed line, which is the critical value for the 95% confidence level, as suggested by Hansen [28]. The confidence interval here was 5–7 h. Furthermore, as a pretesting for the presence of the threshold effect, we carried out Hansen’s testing [29] and obtained a significant *p*-value of 0.032 for the null of no threshold.

Applying a threshold of 6 h of sleep, 116 participants fell into the first subgroup with less than 6 h of sleep per day, and the remaining 295 individuals were in the second subgroup, with 6 or more hours of sleep per day. We refer to the first subgroup as ‘short sleepers’ and the second as the ‘adequate-long sleepers’.

In Table 1, we report mean and standard deviations of the variables included in this study, from both the full cohort and each of the two subgroups. The results of the t-tests and chi-square tests showed that the two groups were very similar with respect to sociodemographic and health-related characteristics, except that there were more women in the short sleeper group. As for health-related variables, there were significant differences between the groups on some measures; for example, the short sleepers had a higher BMI and experienced more symptoms than those of the adequate-long sleeper group. The short sleepers were also more likely to have poorer health-related behavior, in terms of smoking and less engagement in health behaviors, than those of the other group. Short sleepers experienced an average of 4.68 h of sleep per day, whereas adequate-long sleepers slept for an average of 7.32 h daily, with the latter being within the range of the optimal hours of sleep. There was also a significant difference in the average quality of sleep, with short sleepers reporting a lower quality of sleep than adequate-long sleepers.

### 3.3. Effects of Sleep Quality and Duration on Risk of MetS

We first examined the logistic regression results from the full cohort as a benchmark. The logistic regression results (column 1 of Table 2) indicated that some factors, such as sleep quality, were significantly associated with risk of MetS. However, there was no significant association with sleep duration.

We then estimated the log of odds ratio of the risk of MetS for each of the subgroups, which are reported in columns 2 and 3 of Table 2. In these analyses, the effect of sleep quality was statistically significant only for the adequate-long sleeper group. The log of odds ratio was 0.215 (95% CI = 0.07–0.36), which was almost double compared to the log of odds ratio from the full cohort. The odds ratio converted from the log of odds ratio is 1.24 (95% CI = 1.07–1.43). Figure 3 illustrates the effect of sleep quality across the two subgroups. As a graphical summary of this key finding, we calculated the odds ratio of the risk of MetS against sleep quality measures for each group.

### 3.4. Further Analyses

To assess the robustness of the results, we reallocated individuals who reported sleeping 6 h per day (27.5% of participants), which was the threshold point, from the adequate-long sleeper group into the short sleeper group. With this new grouping, the log of odds ratio was 0.345 (95% CI = 0.14–0.55). The effect of sleep quality across the groups remained robust, with no sleep quality effect in the short sleeper group and a significant effect of sleep quality in the adequate-long sleeper group.

We considered the possibility that the effect of sleep quality was affected by sleep duration. The result of a Pearson’s correlation coefficient indicated that sleep quality and hours of sleep were strongly correlated with each other when analysis was conducted on the full cohort (*r* = −0.40, *p* < 0.001). However, once we split the cohort into two based on the threshold, the correlation coefficient was significant only in the short sleeper group (*r* = −0.27, *p* = 0.003), indicating that as hours of sleep increased up to the threshold, the quality of sleep improved for the short sleeper group. In contrast, longer hours of sleep above the threshold did not correspond to quality of sleep. These results indicate that sleep quality in conjunction with sleep duration was an important factor for the risk of MetS, as shown in Table 2 and Figure 3.

## 4. Discussion

We applied a threshold regression approach for a more robust evaluation of the effects of sleep duration and sleep quality on the risk of developing MetS while controlling for sociodemographic and behavioral characteristics that are closely related to health. The estimated threshold point was 6 h of sleep, which corresponds to the lower end of the hours of adequate sleep documented in the literature [8,9]. Our key finding was that there was a distinguishable effect of sleep quality on MetS risk with reference to the estimated threshold of sleep duration (6 h of sleep per day) [9,30]. For short sleepers, classified as individuals who sleep less than 6 h per night, neither sleep duration nor sleep quality was a significant predictor of MetS risk. In contrast, for adequate-long sleepers, whose sleep duration was equal to or above the duration of 6 h, sleep quality was a significant predictor for MetS risk. Therefore, we found an asymmetric interactive effect of sleep duration and sleep quality in predicting MetS risk when examined using a nonlinear framework that incorporates a threshold point.

The results of this study corroborate previous findings that short sleepers tend to show different health conditions compared to adequate-long sleepers [31,32]. However, our study differed from the existing studies in that we considered the association between sleep duration and quality, and we illustrated how the prediction of MetS risk drastically differs depending on whether sleep duration is below or above the threshold (6 h).

The findings of this study also add to the literature in several important ways. First, we found a U-shaped relationship between sleep duration and MetS risk, after controlling for the covariates, indicating that there was a higher risk of MetS in both short and long sleepers compared to adequate sleepers. This is consistent with prior evidence of a U-shaped relationship between sleep duration and CVDs associated with MetS [7,10]. However, once participants were classified by the threshold and sleep quality was controlled for, sleep duration itself was no longer a significant predictor for the MetS risk, which is in contrast to other studies that found both short and long sleep durations are key risk factors of MetS [12,13].

Second, we showed that sleep quality was a significant predictor of MetS only in adequate sleepers, and not in short sleepers, but this heterogeneous effect of sleep quality was not detected when the analysis was conducted on the full cohort, and the threshold point of sleep duration was not taken into account. These findings, therefore, highlight the need to consider both sleep duration and sleep quality when predicting MetS risk.

Third, there was a significant negative correlation between sleep duration and sleep quality in the entire cohort, but the relationship differed across the two subgroups. Specifically, there was a strong negative correlation between sleep duration and sleep quality in short sleepers but this significant correlation was lost in adequate-long sleepers. Again, the fact that there were distinct relationships based on a particular sleep duration serves as evidence of the need to consider both sleep duration and sleep quality in predicting MetS risk.

Lastly, consistent with previous studies, we showed that some sociodemographic factors such as age, income level, and BMI, had significant effects on MetS risk regardless of sleep duration. However, it is noteworthy that health-related behaviors served as a more powerful predictor of MetS risk in short sleepers than in adequate-long sleepers. This finding suggests that active engagement in health-related behaviors should be highlighted to decrease the MetS risk, especially in short sleepers.

In summary, our findings demonstrate that consideration of both sleep quality and quantitative sleep duration are essential for a more accurate estimation of the MetS risk and that threshold estimation is useful for identification of the underlying relationship between sleep duration and sleep quality.

### Strengths and Limitations

Despite previous findings on the roles of sleep duration and sleep quality in predicting the MetS risk, there is limited evidence on their joint effects. Using the threshold estimation and stepwise analysis, we showed that the relationship between sleep duration and sleep quality differs with reference to a specific sleep duration (i.e., 6 h of sleep per day) and that their impacts on MetS risk also differ. Our findings suggest that sleep duration and sleep quality are composite factors that characterize one’s sleep and that threshold estimation is an applicable and useful approach for accurately measuring the joint effects of sleep duration and quality on MetS risk.

In this cross-sectional study, we used convenience sampling: patients were recruited within a limited socio-cultural environment, restricting the generalization of the findings. Future studies using large-scale data are warranted to verify our results. Further, sleep characteristics may vary according to socio-cultural factors such as race and climate. Thus, cross-cultural studies should utilize threshold estimation to substantiate their findings. Lastly, our data were collected through self-reported questionnaires and thus the information collected on sleep quality and duration is subjective. However, numerous empirical studies have found that subjective evaluations on both the quality and quantity of sleep play a critical role in predicting diverse health outcomes, including the risk factors of MetS (e.g., type 2 diabetes, hypertension). An increasing number of sleep research has incorporated the objective features and patterns of sleep derived from smart devices and trackers. It is essential that future studies also include objective data of sleep to capture clearer links with health-related characteristics and thus to specify the physiological mechanisms pertaining to the prediction of MetS risk.

## 5. Conclusions

The impact of sleep quality on the risk of MetS differs with reference to a threshold sleep duration (i.e., 6 h of sleep per day). The interaction between sleep quality and sleep duration was a significant predictor for MetS risk. Stepwise analysis following threshold estimation enabled more accurate analysis of the impact of sleep duration and sleep quality on MetS risk. Future studies should utilize this approach on larger and more diverse study populations to establish further evidence for the relationship between sleep and MetS risk.

## Figures and Tables

**Figure 1 healthcare-08-00186-f001:**
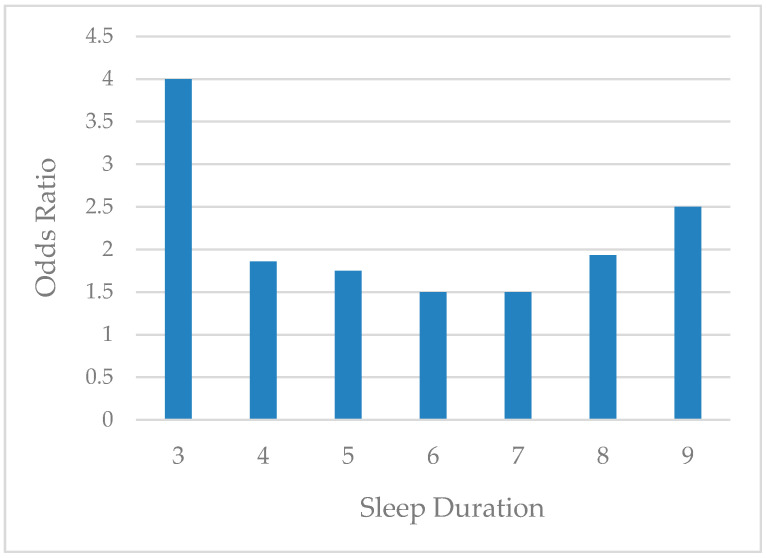
Odds Ratio of the MetS risk by sleep duration. Odds ratio of the MetS risk was plotted against sleep duration in all subjects.

**Figure 2 healthcare-08-00186-f002:**
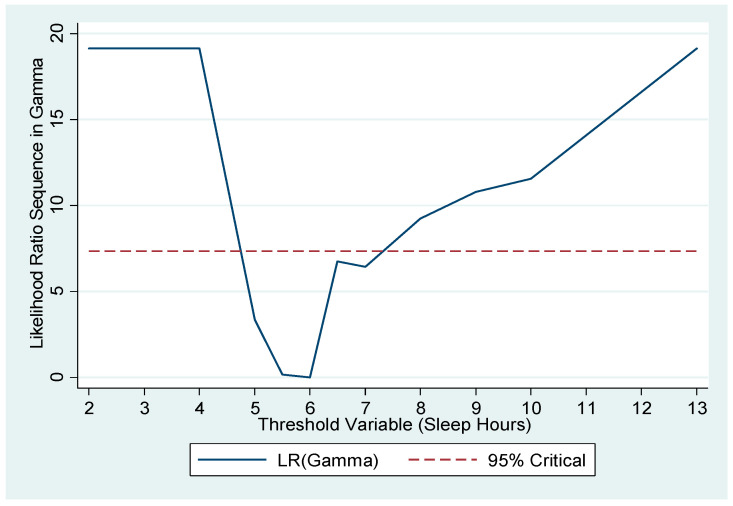
Estimated threshold of sleep duration and its confidence interval. The solid line is the likelihood ratio sequence along different values of the threshold variable (sleep duration) and the dashed line is the critical value for a 95% confidence level. The confidence interval for the threshold is the collection of values below the dashed line.

**Figure 3 healthcare-08-00186-f003:**
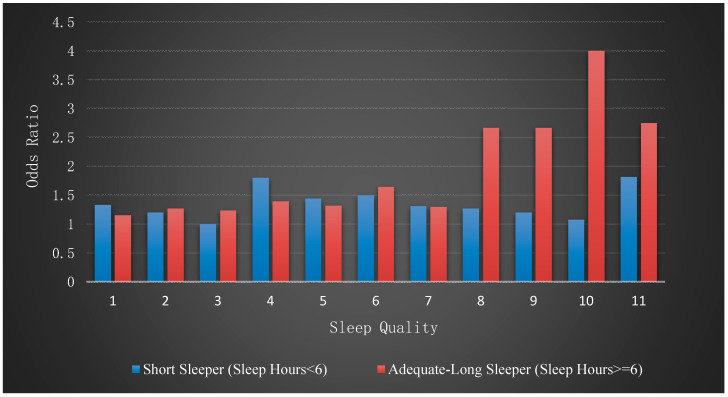
Comparison of odds ratio of risk of MetS with sleep quality for the short sleeper and adequate-long sleeper groups.

**Table 1 healthcare-08-00186-t001:** General characteristics of study participants and comparisons of sleep duration subgroups.

Variables	All		Short Sleeper Group		Adequate-Long Sleeper Group		*t or χ^2^*
	Full Cohort (*N* = 411)		<6 Sleep Hours (*n* = 116)		≥6 Sleep Hours (*n* = 295)		
	M (SD) or *n* (%)		M (SD) or *n* (%)		M (SD) or *n* (%)		
MetS risk ^§^^1^	246	(59.9)	75	(64.7)	171	(58.0)	1.55
Age	52.96	(10.5)	52.32	(10.03)	53.21	(10.66)	−0.78
Sex ^§^^2^	219	(53.3)	52	(44.8)	167	(56.6)	4.64 **
Education ^§3^	212	(51.6)	55	(47.4)	157	(53.2)	1.12
Income level	3.37	(1.34)	3.22	(1.48)	3.37	(1.28)	−1.00
Job ^§^^4^	315	(76.6)	83	(71.6)	232	(78.6)	2.34
BMI	24.60	(3.49)	25.39	(4.04)	24.29	(3.20)	2.93 **
Symptoms	10.34	(5.68)	11.10	(6.01)	10.04	(5.53)	1.72 *
Health behaviors	2.42	(0.48)	2.35	(0.51)	2.45	(0.48)	−1.79 *
Drinker ^§^^5^	193	(47.0)	56	(48.3)	137	(46.4)	0.11
Smoker ^§^^6^	96	(23.4)	36	(31.0)	60	(20.3)	5.32 **
Sleep quality	6.42	(3.77)	9.75	(3.68)	5.11	(2.91)	13.46 ***
Sleep duration	6.57	(1.91)	4.68	(0.69)	7.32	(1.70)	−16.15 ***

Note. ^§^ Dummy variable; Reference groups: ^1^ = no risk factors of MetS; ^2^ = women; ^3^ = education level below university degree; ^4^ = unemployed; ^5^ = no drinking; ^6^ = no smoking. N (n): number of observations. *** *p* < 0.001, ** *p* < 0.05, * *p* < 0.1.

**Table 2 healthcare-08-00186-t002:** Logistic regression analysis for interactive effects of sleep quality and sleep duration on the risk of MetS.

Variables	All		Short Sleeper Group		Adequate-Long Sleeper Group	
	Full Cohort (*N* = 411)		<6 Sleep Hours (*n* = 116)		≥6 Sleep Hours (*n* = 295)	
	Log of Odds Ratio	SE	Log of Odds Ratio	SE	Log of Odds Ratio	SE
Sleep quality	0.116 **	0.051	0.084	0.099	0.215 ***	0.072
Sleep duration	−0.048	0.083	0.168	0.406	−0.171	0.115
Covariates						
Age	0.100 ***	0.021	0.098 **	0.040	0.110 ***	0.026
Sex ^§^^1^	0.091	0.382	1.397	0.873	−0.358	0.468
Education ^§^^2^	0.152	0.344	0.540	0.635	−0.017	0.429
Income level	−0.360 ***	0.133	−0.515 **	0.236	−0.331 *	0.174
Job ^§^^3^	−0.755	0.478	−0.823	0.877	−0.679	0.608
BMI	0.436 ***	0.062	0.291 ***	0.102	0.519 ***	0.082
Symptoms	−0.181 ***	0.030	−0.155 ***	0.059	−0.204 ***	0.037
Health behaviors	−0.785 **	0.329	−1.252 **	0.629	−0.649	0.402
Drinker ^§^^4^	0.153	0.336	0.518	0.645	0.138	0.407
Smoker ^§^^5^	0.566	0.415	−0.774	0.924	0.768	0.501
Constant	−10.304 ***	2.179	−6.918 *	3.608	−12.232 ***	3.006

Note. ^§^ Dummy variable; Reference groups: ^1^ = women; ^2^ = education level below university degree; ^3^ = unemployed; ^4^ = no drinking; ^5^ = no smoking. N (n): number of observations. *** *p* < 0.001, ** *p* < 0.05, * *p* < 0.1.

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
