# Peer review of "Interactive Impact of Sleep Duration and Sleep Quality on the Risk of Developing Metabolic Syndrome in Korean Adults"

_healthcare, 2020, doi:10.3390/healthcare8020186_

Round 1

Reviewer 1 Report

The study employed a novel analysis approach to explore the interactive impact of sleep duration and sleep quality on the risk of developing metabolic syndrome. But some points were not clear and further revisions were needed.

  1. There were some typos, e.g. several words without space in-between, line 112, 138 etc.

Materials and Methods:

  1. How the sample size was determined should be introduced.
  2. In 2.5 statistical analyses: Why the threshold regression was employed? Its advantages in the current study should be introduced clearly. For the regression analysis, which method was used? Enter/forward or backward?

Results

  1. 1 descriptive characteristics of the participants. A table with full introduction of all characteristics was suggested. Detailed data of education, income, MetS variables should be presented.
  2. 2 Based on Figure 2, why the threshold of 6 was selected? Why not 5 or 7? Please  introduced clearly.
  3. Table 1, some variables were not continuous data. It was not appropriate to report the Mean, SD.
  4. How the independent variables were selected? All measured variables were used? Or select some of them? Based on which principle for the selection.
  5. In the regression analyses, for the categorical independent variables like education and income levels, which was the reference group?
  6. Table 2 contains too much information. However, for each regression model, the information was not enough. For each column, meanings of the data should be specified. Are they B, SE, OR or CI?
  7. 4 line 228. the statement of “as hours of sleep increased...quality of sleep became poorer”was wrong. For PSQI, a higher score indicates the poorer sleep quality. Accordingly, some discussion, e.g. line 262, was not appropriate.
  8. If you used PSQI total score as the indicator of sleep quality, it must contains the domain of sleep duration. You could not separate the sleep duration from sleep quality (PSQI total score), unless you used the subjective sleep quality score. Therefore, it is not appropriate to claim “sleep quality itself became an important factor for the risk of MetS”(e.g. line 230).

Author Response

Thank you for your reviews of the manuscript entitled “Interactive impact of sleep duration and sleep quality on the risk of developing metabolic syndrome in Korean adults.” We appreciate the helpful feedback. We carefully addressed your comments and suggestions and incorporated them in the revised version of our paper. For your attention, we present your comments and provide our responses (in italic) in the order in which they appeared in the reviews.

Point-by-point responses to Reviewer 1

1. There were some typos, e.g. several words without space in-between, line 112, 138 etc.

(Response) Thank you for the careful review. We made our best efforts with the help of professional editors to remove any remaining errors in the paper.

Materials and Methods:

2. How the sample size was determined should be introduced.

(Response) We added the information of power analysis using the G*Power program in the sample characteristics on the Materials and Methods section.

“A prior power analysis was conducted to estimate an adequate sample size using G*Power 3.1.9.4 software [21]. Using the type 1 error (α) criterion of 0.05, medium effective size (0.20) for linear multiple regression analysis with 15 predictors, and anticipated power level (.08), the minimum required sample size was calculated to be 108. When considering two groups categorized by threshold point of sleep duration, a minimum number of 216 participants were required. Since the final total number of the participants in the present study was 411 and both groups on the threshold criterion consisted of over 108 participants, the sample size of the present study was adequate.”

3. In 2.5 statistical analyses: Why the threshold regression was employed? Its advantages in the current study should be introduced clearly. For the regression analysis, which method was used? Enter/forward or backward?

(Response) For the estimation of the threshold point, the threshold regression method developed by Lee et al. (2011) was used as described in section 2.5. After the estimation of the threshold of 6, we split the sample by the threshold and used logistic regression for each subsample. To clarify the referee’s concerns, we rewrote the Methods section regarding the threshold regression and logistic regression analyses step by step in section 2.5 as follows.

We employed a threshold regression model to assess the potential asymmetric interaction between sleep quality and sleep duration on the development of MetS. The concept of threshold effect has been widely reported in various fields from biology to toxicology to the medical literature to explain heterogeneous treatment effects due to a covariate. For example, Kamper-Jorgensen and colleagues [25] have reported threshold effect of alcohol intake and cirrhosis, and Hill et al. [26] on threshold effect of exercise intensity and circulating cortisol level. The threshold for sleep duration was determined to maximize the joint logit-likelihood function as in the study by Lee et al. [27]. Precise construction of confidence intervals for the threshold of sleep duration was challenging because hours of sleep, which we will refer to as “sleep duration,” was measured in 30-minute increments in this study and, therefore, was not continuous. Against this backdrop, we report a confidence interval for the threshold sleep duration, as in the study by Hansen [28].

We next split our cohort into two groups, depending on whether sleep duration was above or below the threshold, and conducted t-tests and chi-square tests to examine whether there were differences in sociodemographic and health-related characteristics between the subgroups. Pearson’s correlation coefficients were applied to assess the relationship between sleep quality and sleep duration.

We used logistic regression analyses to examine the effects of various covariate factors, not limited to sleep characteristics, on the risk of MetS for each subgroup divided by the threshold. We provided the estimated coefficients of various risk factors from logistic regression analyses first in the full cohort and then in each of the two subgroups, above or below the threshold, respectively.

Results

4. Descriptive characteristics of the participants. A table with full introduction of all characteristics was suggested. Detailed data of education, income, MetS variables should be presented.

(Response) More details of the variables, especially the categorical variables, were provided in the Results section and in Table 1 following the suggestion.

 “Income levels (average monthly income) were divided into five categories, from 1 (below 1 million Korean won or 830 USD) through 5 (above 4 million Korean won or 3,300 USD), with the average income level being 3, which is analogous to 2.5 million Korean won or 2,075 USD.

The average BMI was 24.6 and about 60% of the cohort was at risk for MetS. The average number of symptoms was 10.34 (SD=5.68, range=0-20) and the mean score of health behavior was 2.42 (SD=0.48, range=0-3). The proportion of current smokers and drinkers was 23.4% and 47.0%, respectively. “

5. Based on Figure 2, why the threshold of 6 was selected? Why not 5 or 7? Please introduced clearly.

(Response) The estimated threshold of 6 is the minimizer of the graph in Figure 2, and a confidence interval of 5 and 7 is the region below the dashed line. In OLS regression, the confidence interval for the coefficient is given by an interval centered at the OLS estimate and two standard deviations. However, for the threshold estimation, we cannot use that approach. Instead, Hansen (2000) suggested a likelihood ratio-based approach as illustrated in Figure 2. Following the referee’s concerns, we explained this point in section 3.2.

6. Table 1, some variables were not continuous data. It was not appropriate to report the Mean, SD.

(Response) Following the suggestion, we reported the frequency for the categorical variables and the mean for the continuous variables.

7. How the independent variables were selected? All measured variables were used? Or select some of them? Based on which principle for the selection.

(Response) We included all variables regarding socioeconomic, health-related, and behavioral characteristics, for which empirical studies have reported their relationships with sleep as covariates. The sample size was large enough to include them for testing.

8. In the regression analyses, for the categorical independent variables like education and income levels, which was the reference group?

(Response) We revised Table 1 to include the information of categorical variables and their reference groups.

9. Table 2 contains too much information. However, for each regression model, the information was not enough. For each column, meanings of the data should be specified. Are they B, SE, OR or CI?

(Response) We added a row at the top to explain that our estimates are log of odds ratio with standard errors in parentheses. We also reported odds ratio (OR) in the text and in Figure 3.

10. 4 line 228. the statement of “as hours of sleep increased...quality of sleep became poorer” was wrong. For PSQI, a higher score indicates the poorer sleep quality. Accordingly, some discussion, e.g. line 262, was not appropriate.

(Response) Thank you for the careful review. We spotted our mistake and corrected the sentence.

“We considered the possibility that the effect of sleep quality was affected by sleep duration. The result of a Pearson’s correlation coefficient indicated that sleep quality and hours of sleep were strongly correlated with each other when analysis was conducted on the full cohort (r= -0.40, p<0.001). However, once we split the cohort into two based on the threshold, the correlation coefficient was significant only in the short sleeper group (r= -0.27, p=0.003), indicating that as hours of sleep increased up to the threshold, the quality of sleep improved for the short sleeper group. In contrast, longer hours of sleep above the threshold did not correspond to quality of sleep. These results indicate that sleep quality in conjunction with sleep duration was an important factor for the risk of MetS, as shown in Table 2 and Figure 3.”

11. If you used PSQI total score as the indicator of sleep quality, it must contains the domain of sleep duration. You could not separate the sleep duration from sleep quality (PSQI total score), unless you used the subjective sleep quality score. Therefore, it is not appropriate to claim “sleep quality itself became an important factor for the risk of MetS”(e.g. line 230).

(Response) As the referee pointed out, the statement can be misleading. The PSQI is the average of sleep duration and other factors. Including both PSQI and sleep duration in the regression is equivalent to running the regression which includes sleep duration and the other factors in PSQI as regressors. It imposes the restriction that the other factors in PSQI have the same coefficient whereas sleep duration can have a different coefficient. Thus, the regression estimates the joint effect of PSQI and sleep duration. Accordingly, we changed the sentence as follows to reflect the reviewer’s concerns.

 “These results indicate that sleep quality in conjunction with sleep duration was an important factor for the risk of MetS, as shown in Table 2 and Figure 3.”

Reviewer 2 Report

The aim of the study is to examine the impact of sleep duration and sleep quality on the risk of developing metabolic syndrome. While in results section its' clearly presented that both- short sleepers and adequate to long sleepers have similar MetS risk (see table 1) and sleep quality increases MetS risk in total and adequate to long sleepers groups (see table 2), not in the short sleepers group, the conclusion "both, sleep duration and sleep quality are crucial determinants of the development of metabolic syndrome" is faulty and misinterpret. The results section 3.4 is speculative and does not support the conclusions. Therefore results and conclusion sections must be substantially overworked and revised. That is my major concern.

As conclusions are based entirely on subjective data, it should be discussed and clearly stated as limitation of the study.

Author Response

Thank you for your reviews of the manuscript entitled “Interactive impact of sleep duration and sleep quality on the risk of developing metabolic syndrome in Korean adults.” We appreciate the helpful feedback. We carefully addressed your comments and suggestions and incorporated them in the revised version of our paper. For your attention, we present your comments and provide our responses (in italic) in the order in which they appeared in the reviews.

Point-by-point responses to Reviewer 2

  1. The aim of the study is to examine the impact of sleep duration and sleep quality on the risk of developing metabolic syndrome. While in results section its' clearly presented that both- short sleepers and adequate to long sleepers have similar MetS risk (see table 1) and sleep quality increases MetS risk in total and adequate to long sleepers groups (see table 2), not in the short sleepers group, the conclusion "both, sleep duration and sleep quality are crucial determinants of the development of metabolic syndrome" is faulty and misinterpret. The results section 3.4 is speculative and does not support the conclusions. Therefore results and conclusion sections must be substantially overworked and revised. That is my major concern.

(Response) As the referee pointed out, we presented in Table 1 that the average MetS risk is similar between short and adequate-long sleepers. However, when we examined the effect of sleep quality on the MetS risk using the model where we took into account the various individual characteristics that were listed in Table 1, we found that sleep quality has a different effect depending on sleep hours.

 As described in the paper, we used a threshold estimation approach and found that the effect of sleep quality on the MetS risk was different depending on the threshold point of “sleep duration-6 hours.” However, we also found that both the sleep quality and sleep duration were not significant predictors on the MetS risk in the full sample. The findings demonstrate that the consideration of sleep duration is necessary to investigate the effect of sleep quality on the MetS risk. That is why we described that "both, sleep duration and sleep quality are crucial determinants of the development of metabolic syndrome." This heterogeneous effect of sleep quality on the MetS risk can be captured by the threshold regression model and we described this approach in Methods section 2.5 following the other referee’s suggestion.

 “For further assessment of potential asymmetric interaction between sleep quality and sleep duration on the development of MetS, we employed a threshold regression model. The concept of threshold effect has been widely reported in various fields from biology to toxicology to the medical literature to explain heterogeneous treatment effects due to a covariate. For example, Kamper-Jorgensen and colleagues [25] have reported threshold effect of alcohol intake and cirrhosis, and Hill et al. [26] on threshold effect of exercise intensity and circulating cortisol level. The threshold for sleep duration was determined to maximize the joint logit-likelihood function as in the study by Lee et al. [27]. Precise construction of confidence intervals for the threshold of sleep duration was challenging because hours of sleep, which we will refer to as “sleep duration,” was measured in 30-minute increments in this study and, therefore, was not continuous. Against this backdrop, we report a confidence interval for the threshold sleep duration, as in the study by Hansen [28].”

 (Response continued) We also rewrote the Results section 3.4 to resolve the referee’s concerns.

“We considered the possibility that the effect of sleep quality was affected by sleep duration. The result of a Pearson’s correlation coefficient indicated that sleep quality and hours of sleep were strongly correlated with each other when analysis was conducted on the full cohort (r= -0.40, p<0.001). However, once we split the cohort into two based on the threshold, the correlation coefficient was significant only in the short sleeper group (r= -0.27, p=0.003), indicating that as hours of sleep increased up to the threshold, the quality of sleep improved for the short sleeper group. In contrast, longer hours of sleep above the threshold did not correspond to quality of sleep. These results indicate that sleep quality in conjunction with sleep duration was an important factor for the risk of MetS, as shown in Table 2 and Figure 3.”

  1. As conclusions are based entirely on subjective data, it should be discussed and clearly stated as limitation of the study.

(Response) We agree with your comments. We addressed the limitations with regard to the subjective data in the Discussion section.

“Lastly, our data were collected through self-reported questionnaires and thus the information collected on sleep quality and duration is subjective. However, numerous empirical studies have found that subjective evaluations on both the quality and quantity of sleep play a critical role in predicting diverse health outcomes including the risk factors of MetS (e.g., type 2 diabetes, hypertension). An increasing number of sleep research studies has incorporated the objective features and patterns of sleep derived from smart devices and trackers. It is essential that future studies also include objective data of sleep to capture clearer links with health-related characteristics and thereby specify the physiological mechanisms pertaining to the prediction of MetS risk.”

Reviewer 3 Report

The concept of the manuscript is relevant. However, I find several flaws in the experimental design. Specifically, there are issues with choice of variables and interpretation of data as specified below - 

The most critical issue of this manuscript is the choice of PSQI as a variable. It is known that PSQI is a subjective measure of sleep. Moreover, PSQI is also not very reliable in case of older adults. The study group includes subjects up to the age of 84. Therefore, in my opinion, PSQI doesn't reflect the correct results.

Secondly, the presence of huge age variation in the study group makes the interpretation of result little difficult. It is well known that age affects sleep. How the authors address that issue?

Furthermore, the analysis of threshold point of sleep was performed using the complete data. It is highly likely that depending on age/sex and other covariates, the threshold point can change. Have the authors corrected for that?

Since I find that the above issues are not sufficiently addressed here, the manuscript may be considered after a major revision.

Author Response

Thank you for your reviews of the manuscript entitled “Interactive impact of sleep duration and sleep quality on the risk of developing metabolic syndrome in Korean adults.” We appreciate the helpful feedback. We carefully addressed your comments and suggestions and incorporated them in the revised version of our paper. For your attention, we present your comments and provide our responses (in italic) in the order in which they appeared in the reviews.

Point-by-point responses to Reviewer 3

The concept of the manuscript is relevant. However, I find several flaws in the experimental design. Specifically, there are issues with choice of variables and interpretation of data as specified below – 

  1. The most critical issue of this manuscript is the choice of PSQI as a variable. It is known that PSQI is a subjective measure of sleep. Moreover, PSQI is also not very reliable in case of older adults. The study group includes subjects up to the age of 84. Therefore, in my opinion, PSQI doesn't reflect the correct results.

(Response) We understand your concern that the PSQI may not be reliable in older persons since it is a subjective measure. However, we believe that this reliability issue on subjective measurement is not limited only to older people but applies to all ages of the sample. Internal consistency for the PSQI in this study was adequate (Cronbach’s α =.86). Further, we controlled for the effect of age along with other sociodemographic, health-related, and behavioral characteristics when estimating the effects of sleep quality and hours on the MetS risk. Nevertheless, we addressed the limitation of our study as it is based on subjective measures in the Discussion section as the other referee suggested.

  1. Secondly, the presence of huge age variation in the study group makes the interpretation of result little difficult. It is well known that age affects sleep. How the authors address that issue? Furthermore, the analysis of threshold point of sleep was performed using the complete data. It is highly likely that depending on age/sex and other covariates, the threshold point can change. Have the authors corrected for that? Since I find that the above issues are not sufficiently addressed here, the manuscript may be considered after a major revision.

(Response) As the referee pointed out, other variables, such as age, sex, and health behaviors could affect the threshold point. We considered these individual characteristics as covariates in detecting the threshold point of sleep duration. We added this information in the Results section for a clearer description. We thank the referee for this suggestion.

 “In the estimation of the threshold point, we controlled for individual characteristics including age, gender, education, and other variables that are listed in Table 1.”

(Response continued) We examined the effects of sleep quality and duration in all of the samples as well as in each of the age groups by including age or splitting the sample by age group in the model. We found that the threshold point was consistent with the confidence interval for each age group that fell between 5 and 7 as in the full sample. For a graphical illustration, we presented the estimated threshold point in each age group, 40’s, 50’s, and 60’s-70’s.

We also estimated the sleep quality and sleep duration effects on the MetS risk by focusing on individuals aged 40 to 79. We found similar findings as in the prior results from the full sample. We presented the estimation results from this age-restricted sample in the subsequent pages.

Overall, the empirical evidence of a threshold point of 6 hours and different effects of sleep quality depending on sleep hours, even in a diverse population with a broad range of ages, provides more confirming evidence for the effects of sleep quality and quantity in predicting the MetS risk.

The threshold of sleep hours is estimated by age group. The figures below illustrate the estimated threshold between 6 and 7 with confidence interval between 5 and 7, consistent with our results presented in the paper, and the findings are robust across age groups.

Please see the attached file for additional Figures and Table. 

Figure A. The threshold estimate for the age 40 group is 6 with CI [6,6]

Figure B. The threshold estimate for the age 50 group is 7 with CI [5,7]

Figure C. The threshold estimate for the age 60-70 group is 6 with CI [5,6]

Table. The effect of sleep quality on development of MetS risk differs depending on whether an individual is in the short sleeper or adequate-long sleeper group.

Estimated Risk Factors of Metabolic Syndrome

(1)

(2)

(3)

All

Short sleeper

Adequate-long sleeper

VARIABLES

Full sample

Sleep hours<6

Sleep hours>=6

Odds ratio (SE)

Odds ratio (SE)

Odds ratio (SE)

Sleep quality

0.109**

0.129

0.192**

(0.053)

(0.121)

(0.075)

Sleep duration

-0.008

0.399

-0.154

(0.087)

(0.489)

(0.119)

Age

0.151***

0.200***

0.153***

(0.027)

(0.061)

(0.032)

Gender

-0.005

1.806*

-0.510

(0.407)

(0.962)

(0.495)

Education

0.183

0.565

0.057

(0.363)

(0.719)

(0.443)

Income level

-0.321**

-0.541**

-0.274

(0.141)

(0.270)

(0.183)

Job

-0.605

-0.871

-0.437

(0.509)

(1.031)

(0.640)

BMI

0.436***

0.261*

0.499***

(0.067)

(0.139)

(0.085)

Symptoms

-0.171***

-0.128*

-0.196***

(0.032)

(0.066)

(0.039)

Health Behaviors

-0.876**

-1.308*

-0.800*

(0.354)

(0.706)

(0.432)

Drinker

0.217

1.322*

0.008

(0.354)

(0.737)

(0.424)

Smoker

0.783*

-1.076

0.991*

(0.429)

(1.031)

(0.512)

Constant

-13.481***

-13.777***

-14.114***

(2.484)

(4.829)

(3.315)

Observations

385

104

281

*** p<0.01, ** p<0.05, * p<0.1

Round 2

Reviewer 1 Report

The variables in Table 1 should be marked clearly, indicating they were presented in Mean (SD) or n (%).